# The Motivational Underpinnings of Intentions to Use Doping in Sport: A Sample of Young Non-Professional Athletes

**DOI:** 10.3390/ijerph18105411

**Published:** 2021-05-19

**Authors:** Andrea Chirico, Fabio Lucidi, Gennaro Pica, Daniela Di Santo, Federica Galli, Fabio Alivernini, Luca Mallia, Arnaldo Zelli, Arie W. Kruglanski, Antonio Pierro

**Affiliations:** 1Department of Psychology of Development and Socialisation Processes, “Sapienza” University of Rome, 00185 Rome, Italy; fabio.lucidi@uniroma1.it (F.L.); daniela.disanto@uniroma1.it (D.D.S.); federica.galli@uniroma1.it (F.G.); fabio.alivernini@uniroma1.it (F.A.); antonio.pierro@uniroma1.it (A.P.); 2Law School, University of Camerino UNICAM, 62032 Macerata, Italy; gennaro.pica@unicam.it; 3Department of Movement, Human and Health Sciences, University of Rome “Foro Italico”, 00135 Rome, Italy; luca.mallia@uniroma4.it (L.M.); arnaldo.zelli@uniroma4.it (A.Z.); 4Department of Psychology, University of Maryland, College Park, MD 20742, USA; hannahk@umd.edu

**Keywords:** doping intentions, significance loss, passion, normative influence, moral disengagement

## Abstract

Doping use is considered as a deviant behavior in sport contexts, and it is necessary to recognize preventive factors to shut down the negative consequences. We proposed that athletes experiencing loss of personal significance would be more prone to doping use intentions. This pathway should occur through the effect of the enhanced predominance of obsessive (vs. harmonious) passion that such athletes experience concerning their sport activity, which, in turn, facilitates the adoption of moral disengagement strategies to find justifications for it, when they perceive that significant others approve their intention. The study relied on a cross-over design, with a convenience sample of 437 athletes recruited at four sports sciences Universities evenly distributed in Italy. Questionnaires administered contained a validated tool based on Kruglanski’s theorizing on radical and deviant behavior (e.g., Loss of Significance, Obsessive, and Harmonious passion) and deriving from social cognitive theory (e.g., Moral disengagement). Results of the study tested a serial mediation moderated model, which links the different variables to explain the influence they have on the intentions to use doping. Overall, this research suggests a motivational dynamic that may be at the heart of illicit behaviors in sport, such as using drugs-enhancing performance potentially among athletes of all kinds.

## 1. Introduction

The use of doping is recognized as a relevant issue in sport. Doping is sometimes considered (even though only tacitly) a means for goal attainment given its relative diffusion not only among professionals but also among non-professionals and even adolescents [1,2,3,4]. The question is, why? What exactly drives this dangerous and deviant behavior to shut down possible negative consequences as preventive factors? We argue that viewing doping only as a means of “winning” may fail to consider less recognizable motivations behind such behavior. Some risk behaviors are engaged in when perceived as relevant to the person’s motivation, in other words, they may represent a person’s attempt to fulfill current motivations or needs, as scholars have previously suggested [5]. Deepening this view, doping behavior can also be examined as a means of satisfying more essential needs of the person. The present research aims to address this point by adopting a novel conceptual framework of radical and deviant behavior, namely the Significance Quest Theory (SQT) [6,7,8]. As far as we know, this is the first attempt to apply the SQT framework to doping behaviors in sports.

In line with SQT, we suggest that doping behavior has unique characteristics that might be attractive for some individuals. Those who have a strong need to fill a void in their life and to (re)gain personal significance (i.e., a fundamental need to count, to be recognized and respected), for whatever reason, are more prone to allocate all their personal resources to their sportive activities even at the cost of other pursuits. This is because these individuals are more likely to become obsessed with their sportive activities (i.e., obsessive passion) [9]. This motivation might lead people to the use, if available, of illegitimate means, such as doping, in order to attain their focal goals [10]. Moreover, we suggest that moral disengagement strategies might be the mechanisms by which these individuals justify and legitimate to themselves the use of doping substances, especially when they perceive that significant others of their social network corroborate their choice.

In what follows, we first introduce the theoretical background of the present research, and then we will more thoroughly present our hypotheses.

### 1.1. Doping Behavior

Doping is referred to as the assumption of drugs that are intended to improve physical performance. The use of these substances is illegal for two main reasons: (1) It is harmful and dangerous for athletes’ health; and (2) it threatens the moral integrity of sports; it is not fair. Although doping may preclude athletes from their careers even for life, drug use in sport and exercise settings is on the rise and calls for deeper understanding and solutions. In this vein, psychological research has given some hints about the determinants of the use of Performance Enhancing Substances (PES). For instance, it has been largely shown that attitudes towards doping, subjective norms, and behavioral control (or self-efficacy), all well predict doping intention and behavior [2,11,12,13]. Consistently, research has shown that doping intention is reduced by anticipated regret [14,15], personal norms, meaning that non-users held more restrictive norms, and is promoted by perceived use by others [16]. The latter finding is consistent with research that has applied the prisoner’s dilemma to the doping situation by showing that professional athletes are more likely to use PES when they perceive their competitors as unfair (i.e., capable of the use of PES as well) [17]. Furthermore, whereas autonomous motivation (i.e., motivation resulting from enjoyment or personal value) and mastery-oriented achievement goals (i.e., emphasis on personal improvement and effort) reduce doping intention, controlled motivation (i.e., motivation resulting from pressure, social approval, or feelings of guilt) and performance-oriented achievement goals (i.e., emphasis on performance as a meter of social comparison) does not [18]. Another important factor that may induce the use of PES for professional athletes is the pressure they feel from trainers, family, friends, fans, the greater the pressure, the greater the likelihood of taking illegal paths (if necessary) in order not to disregard their expectations [19].

Importantly, Ehrnborg and Rosén [19] have also suggested that there are several motives behind PES use that slightly differ among professional and non-professional athletes. Among non-professional athletes (e.g., common people in the gym), doping may serve goals such as appearance and physical toughness, and power [20]. This would probably give them positive attention among friends and (potential) partners, finally resulting in a sense of self-confidence and well-being [19]. Among professional athletes, things change in that the main driving force here is the urge to win. Ehrnborg and Rosén [19] noted that achieving victories or great performances meet the need for self-achievement (i.e., realize one’s ambition) and give athletes honor and glory. For instance, at the highest levels, the winner may not only earn a considerable amount of money but also may quickly achieve worldwide visibility.

Consistent with this viewpoint, we propose that behind the desire for winning, appearance or social recognition is a strong desire to fill a void, which often originates from people’s feelings that their life is worthless and meaningless [21]. Scholars have called this concept the *loss of personal significance* [22,23], which is described next.

### 1.2. Loss of Personal Significance

The theory of quest for significance posits that the fundamental desire to matter, to be respected, and to feel meaningful is a major driving force in human affairs [6,7,8,22,24]. Individuals prompted by this need are more likely to commit fully to important goals, which are particularly suited to enhance and/or restoring their sense of meaning and mattering [25]. As a consequence, people are more likely to use “whatever means they perceive as necessary, regardless of how extreme those means are” [26] (p. 818). An intense quest for significance does not lead to extreme actions directly, but it boosts a person’s readiness to tolerate and enact them for the sake of significance and dignity.

Importantly, however, the quest for significance must first be awakened before it can drive behavior. According to the theory [6], this need can be triggered by the experience of *significant loss* through, for instance, experiences of humiliation and failure. When people suffer such a loss, they desperately seek to regain significance, dignity, and respect [22]. The stronger the lack of personal significance, the greater the motivation to restore a sense of worth and meaning [26]. This is because when a need strongly overcomes the others in strength, these are suppressed, and motivational imbalance occurs [10]. Motivational imbalance frees the individual from the constraints deriving from other needs, allowing for the implementation of the most instrumental means, one most likely to produce goal fulfillment [10]. Thus, these individuals may view the use of doping substances as a particularly effective way of gratifying their dominant concern, despite the cost it exacts. This characteristic is known as *counterfinality*, meaning that a means serves a focal goal (e.g., winning) while undermining other goals (e.g., health, etc.) [5,23,27]. Research has shown that, paradoxically, the greater the perceived cost associated with a *counterfinal* means, the more people perceive it as *instrumental* to the goal it purportedly serves [23,28]. This is because people generally believe in a “no pain, no gain” heuristic that becomes especially salient when reaching one’s goal seems unlikely [23,28,29].

Given that, the question is why people with a strong need to restore their personal significance should, indeed, be more prone to inhibit constraints deriving from other goals and use doping substances? Our answer is the type of passion individuals develop for their sportive activity. Significance seekers can obsessively devote themselves to something, automatically neglecting other life domains [21]. Accordingly, recent research found that the need for significance search is positively related to *obsessive passion* for an activity dear to the person [30]. This concept is described below.

### 1.3. Passion

Passion is defined as a strong inclination towards an activity that people find important and in which they invest a significant amount of time and energy [9]. Vallerand et al. in 2003 [9] proposed a dualistic model that posits the existence of two types of passion: obsessive and harmonious. Obsessive passion (OP) refers to a controlled internalization of the activity into one’s own identity that places a strong and uncontrollable urge to engage in that activity. Such pressure makes it difficult for the person to completely disengage from thoughts about the activity, which overwhelms attention and consumes mental resources, becoming difficult to regulate and integrate it with other life domains. Harmonious passion (HP) refers to an autonomous internalization of the activity into the person’s identity. People feel the passionate activity as an important, but not overwhelming, part of their life, well integrated with other domains, and feel no obligation to engage in it, but rather freely choose to do so [9].

Although research has mainly focused on the unique effects of both passion dimensions, the dualistic model outlines that the two are relatively independent of each other, meaning that people can experience different degrees of *both* HP and OP toward the same activity [31]. This means that similar levels of both HP and OP can be experienced, but one of the two passions can become predominant over the other by showing its *pure* aspect (i.e., where one is high and the other is low). In this latter case, people are more likely to act and experience outcomes related to the predominant passion at that time [31,32]. According to this *within-person differences in construct dimensions* approach, in the present research, we focused our analysis on the predominance type of passion, specifically on the predominance of OP over HP that in previous research was generally associated with more negative outcomes [31] and with radical and deviant behavior [32].

Importantly, OP, as opposed to HP, correlates with indices of psychological maladjustment [9,33] even in sports contexts [34,35,36]. OP, as opposed to HP, also leads to a more rigid and conflicted form of engagement in risky behaviors, such as winter cycling on icy roads [9] or dancing while injured [37]. Given the experience of conflict between the passionate activity and other life domains, OP, as opposed to HP, is also related to greater suppression of goals (i.e., goal shielding) [32,38]. Bélanger and colleagues (2019) also show that people who obsessively engage in the passionate activity tend to favor counterfinal means. Accordingly, athletes obsessively passionate about their sport display more permissive attitudes towards PES, as opposed to harmoniously passionate athletes, who see sport as a more balanced addition to their life [39].

Suppression of alternative goals in OP [32,40] can also involve moral aspects; indeed, a recent study [40] showed that OP facilitates the deactivation of moral self-regulatory processes. In line with these findings, and with the idea of catching up the *pure* aspect of HP or OP [31], we expect people’s predominance of obsessive (vs. harmonious) passion for their sport to positively predispose them to moral disengagement, especially when people’s social network would approve their behavior. Before discussing the construct of moral disengagement in the next section, two words need to be said about people’s social networks and their functions. Social networks (friends, family, acquaintances, etc.) may influence people’s behavior in two ways called (1) *informational influence* and (2) *normative influence* [7,8,22]. First, the network can exert its effects by validating a given action as legitimate (i.e., the agreement with others as evidence of action’s veracity and goodness; *Informational Influence*). Second, the network exerts its effects by rewarding individuals for enacting a given behavior (i.e., others’ approval and agreement as the desired outcome; *Normative Influence*). We believe that the normative influence function of social networks may play a significant role in doping behavior by also promoting, especially for predominant obsessive athletes, the adoption of moral disengagement strategies for freeing behavior from restrictions. This idea is consistent with the findings of Lucidi et al. [2], showing that subjective norms (i.e., the extent to which significant others are perceived as being in accordance with one’s behavior), indeed, positively predict moral disengagement.

### 1.4. Moral Disengagement

Moral disengagement [41,42] is the cognitive process by which one can deactivate moral self-regulation and disengage from moral norms without apparent guilt or self-censure. In general, moral disengagement mechanisms allow people to mitigate the moral consequences of harmful acts by reformulating their causes and consequences [42,43]. For this purpose, people can use moral justification (e.g., portraying unethical behavior as aimed at social or moral purposes), euphemistic labeling (e.g., disguising harmful behavior as if it was acceptable or benign), advantageous comparisons (e.g., comparing unethical behavior with worse conduct), displacement of the responsibility (e.g., viewing their actions as dictates of legitimate authority or social pressure), diffusion of responsibility (e.g., diffusing the responsibility for a joint action by feeling that the other members of the group are equally responsible), disregard or misrepresentation of harmful consequences (e.g., minimizing the impact of their actions), attribution of blame (e.g., by considering victims responsible for their condition), and dehumanization (e.g., viewing victims as lesser beings and unworthy of empathy) [42].

A large body of empirical findings confirms the role of moral disengagement in the expression of immoral and/or harmful behaviors, such as aggression and violence [44], counterproductive behavior in the workplace [45], academic cheating [46], and civic duties violations, such as driving violations [47]. Importantly, even in the sporting context, the use of moral disengagement mechanisms shows significant effects. Moral disengagement has positively predicted attitudes towards PES in competitive athletes [48] and the propensity to use doping substances in adolescents [4,49].

### 1.5. The Present Research

The aim of the present research is to investigate the motivational underpinnings of intentions to use doping substances. Based on recent theorizing on radical and deviant action [6,7,8], we suggest that for the individuals’ experiencing loss of personal significance (and the subsequent need to restore it), the doping possibility is promoted by their enhanced obsession with their sportive activities, which, in turn, facilitates the adoption of moral disengagement strategies to find justifications for it, particularly when they perceive that significant others of their social network would approve and encourage their choice (i.e., when normative influence promote the choice to use PES). This last point is of particular importance as the person’s social network may influence their behavior through its power to reward the individual with approval and consensus [6,7,8]. Athlete’s social networks can influence (both tacitly or expressly) the intention to engage in doping behavior by validating the importance of the goal (i.e., winning) and by confirming that illegal means are legitimated way to pursue it. When doping is perceived to be socially validated and approved, people experience less guilt and distress than when it is questioned by important others. This is confirmed by research showing that subjective norms influence doping behavior [2,11,12,13].

In sum, we aim to test a serial moderated mediation model whereby loss of personal significance (*x*) predicts doping intentions (*y*) because it enhances predominance in obsessive (vs. harmonious) passion (*m*_1_), which, in turn, predicts the adoption of moral disengagement strategies (*m*_2_), this last linkage being moderated by the normative influence stemming from the athletes’ social network (*w*), in such a way as to be stronger in the condition of high (vs. low) normative influence. This model is graphically shown in Figure 1.

## 2. Materials and Methods

### 2.1. Participants and Procedures

Students were contacted through announcements on the notice boards of the universities involved. We used a convenience sampling procedure, and they voluntarily decided to participate in this research. All the students who required information were accepted to participate in the study.

### 2.2. Research Design

The design of the study was a cross-sectional research approach.

### 2.3. Ethics

The study was approved by the Ethics Review Board of the Department of Social and Developmental Psychology, “Sapienza” University of Rome. The study relied on written consent for participation. Students distributed evenly across 4 Italian university degree programs in sport sciences (i.e., Rome, Verona, Chieti, and Cassino) were initially contacted and fully informed about the general aims of the study, then written informed consent was collected.

The students were asked to answer a questionnaire including the measures described in the next section.

### 2.4. Measures

*Loss of significance (LoS)* was measured as participants’ personal feelings of humiliation and shame [50]. Participants responded to 5 items assessing the frequency with which they experienced feelings of “humiliation”, “shame”, “excluded”, “emarginated”, and “people laughing at them” in their daily life. Responses were provided using a 5-point scale (1 = “*Rarely or never*”; 5 = “*Very often*”). Internal consistency analyses supported the reliability of the loss of significance measure (*α* = 0.90).

*Passion for sport* was assessed with the Italian version of the Passion for Sport Scale [35] developed by Vallerand and colleagues [9], which was composed of 2 subscales assessing harmonious (6 items, e.g., “My sport is in harmony with other activities in my life”) and obsessive passion (6 items, e.g., “I have difficulties controlling my urge to do my sport”). Students were asked to express their agreement with each item on a 7-point Likert scale ranging from 1 (“*Do not agree at all*”) to 7 (“*Completely agree*”). Internal consistency analyses supported the reliability of both the harmonious passion (*α* = 0.82) and obsessive passion (*α* = 0.89) subscales. Furthermore, following previous literature [31,32], we calculated a single continuous measure as the predominance of Obsessive over Harmonious passion. We followed Higgins, Pierro, and Kruglanski’s method for scoring (i.e., obsessive passion scores minus harmonious passion scores), which indicated the predominant type of passion of the participants [32,51]. Analyses were based on this difference score, for which a higher score indicated stronger obsessive passion predominance.

*Moral disengagement beliefs* were assessed through a 6 items scale developed and validated in previous studies [2]. Each of these 6 items referred to one of the following moral disengagement mechanism: (1) Advantageous comparison (Item: “Compared to the damaging effects of alcohol and tobacco, the use of illicit substances is not so bad”); (2) diffusion of responsibility (Item: “It is not right to condemn those who use illicit substances to improve their body, since many do so”); (3) euphemistic labeling (Item: “Doping is just a way to maximize your potential”); (4) displacement of responsibility (Item: “Those who take banned substances in sport are not at fault, the fault lies with those who expect too much from him/her”); (5) distorting consequences (Item: “There is no reason to punish those who use banned substances to improve their physical appearance, basically it does not hurt anyone”); (6) moral justification: (Item: “To overcome one’s limits, it is also permissible to use prohibited substances”). No items assessed the 2 mechanisms concerning victims (i.e., “attribution of blame” and “dehumanization”), which were rarely advocated in doping contexts. For each item, students rated their agreement on a 5-point scale ranging from 1 (“*I do not agree at all*”) to 5 (“*I completely agree*”). The 6 items were averaged to create a composite score of moral disengagement (*α* = 0.85).

*Normative Influence* (of athletes’ social network) was assessed through the following 2 items derived from subjective norms scale already used in other similar studies [2] and measuring the perception that the significant others would approve the use of performance enhancing substances: (1) “Think about the people you consider significant in your life: How much would they approve you taking products over the next months to improve your sports performance or physical appearance?”; (2) “How much do you think they would encourage you to take products over the next months to improve your sports performance or physical appearance?”. Responses were recorded on a 5-point Likert scale ranging from 1 (e.g., “*They would not approve at all*”) to 5 (e.g., “*They would definitely approve*”). The 2 items were averaged to create a composite score of normative influence of social network (*α* = 0.84).

*Intentions to use doping* were assessed through the following three separate items [2] measuring the likelihood of using doping substances in the next months (i.e., “How strong is your intention to use illegal substances to improve your sport performance or your physical appearance in the next months?”; “What is the probability that you will use illegal substances to improve your sport performance or your physical appearance in the next months?”; “Do you think you will decide to use illegal substances to improve your sports or appearance performances in the next months?”) Responses were recorded on a 5-point Likert scale ranging from 1 (“*Not at all strong/likely*”) to 5 (“*Very strong/likely*”). The 3 items were averaged to create a composite score of intentions to use doping (*α* = 0.81).

### 2.5. Data Analysis

The data analysis was performed using SPSS (IBM SPSS Statistics for Windows, Version 26.0. Armonk, NY, USA: IBM Corp). The reliability of the variables was tested using Cronbach’s alpha (*α*; Cronbach, 1951). Reliability was considered “excellent” for values of Cronbach’s *α* ≥ 0.90, “good” for α between 0.90 and 0.80 and “acceptable” for α between 0.80 and 0.70 (Kline, 2013). Pearson correlation coefficients were used to examine associations between the study variables.

The mediation model analysis was tested according to Hayes (2018) recommendations. A serial mediation moderated model was tested (model 91, PROCESS) with 5000 bootstrap samples and unstandardized regression coefficients. In the serial mediation analysis, the variables presumed as causally prior were modeled as affecting all variables later in the causal sequence. According to Hayes (2018), this was the most complex serial mediator model possible because it maximized the number of paths that needed to be estimated. Our hypothesized model (Figure 1) was a 2-mediator model in which Loss of Significance was modeled as affecting Doping intention through 4 pathways. One pathway was indirect and ran from LoS to Doping intentions through the first mediator (i.e., Predominance of Obsessive over Harmonious Passion) only, a second indirect path ran through the second mediator (i.e., Moral Disengagement) only, and a third indirect influence passed through both the 2 mediators in serial, with the first affecting the second. The remaining effect of LoS was direct from the dependent variable (i.e., Doping intentions) without passing through the mediators. Furthermore, we evaluated the role of Normative Influence of social network as moderator between the Obsessive-Harmonious Passion and the Moral Disengagement Mechanism, using a mean centering approach, the interaction term between the two was based on these centered scores.

The significance of the mediating and moderating effects was ascertained using bootstrap procedures with 5000 samples, following recent recommendations [52]. Significant indirect effects were evaluated trough the confidence interval, specifically, an effect is considered significant if it does not include 0.

## 3. Results

Recruitment procedures led to the actual rate of 437 university sport sciences students. The sample was composed by 46.9% of females (mean age = 22.7, SD = 2.85), they practiced fitness activities (33.4%), individual (32.9%) and team sports (33.6%).

Descriptive statistics and correlations between the key variables of the study are reported in Table 1.

Results showed that Loss of Significance was positively and significantly correlated with the predominance of Obsessive passion over the Harmonious one (*r* = 0.219, *p* < 0.001). This latter was also positively and significantly correlated with Moral Disengagement (*r* = 0.225, *p* < 0.001), Normative Influence of social network (*r* = 0.122, *p* < 0.05), and Doping Intentions (*r* = 0.226, *p* < 0.001). Moral Disengagement was significantly and strongly related to Normative Influence (*r* = 0.408, *p* < 0.001) and Doping Intentions. (*r* = 0.483, *p* < 0.001). Finally, Normative Influence, was significantly and strongly related to Doping Intentions (*r* = 0.560, *p* < 0.001).

### Serial Moderated Mediation Model

The serial moderated mediation model analysis was tested according to Hayes recommendations [53]. Specifically, results of this model are summarized in Figure 2. As can be seen in the Figure 2 and in accordance with our hypotheses, loss of personal significance positively and significantly predicted predominance in obsessive (vs. harmonious) passion (*m*_1_), *b* = 0.20, *SE* = 0.04, 95% CI [0.117, 0.286], *p* < 0.001, which, in turn, positively and significantly predicts the adoption of moral disengagement strategies (*m*_2_), *b* = 0.09, *SE* = 0.02, 95% CI [0.044, 0.143], *p* < 0.001. Subsequently, moral disengagement strategies positively and significantly predict intentions to use doping (*y*), *b* = 0.45, *SE* = 0.04, 95% CI [0.364, 0.529], *p* < 0.001.

Importantly to note is that the normative influence of social network, besides exerting a main effect on moral disengagement (*b* = 0.37, *SE* = 0.05, 95% CI [0.276, 0.463], *p* < 0.001), it exerts an interactive effect with the predominance in obsessive passion on moral disengagement strategies (*b* = 0.13, *SE* = 0.03, 95% CI [0.064, 0.194], *p* < 0.001), thus that the conditional direct effect of predominance in obsessive passion on moral disengagement strategies was strengthened in the high (1 SD above the mean) normative influence condition (*b* = 0.18, *SE* = 0.03, 95% CI [0.117, 0.245], *p* < 0.001). On the contrary, in the low (1 *SD* below the mean) normative influence condition this effect became non-significant (*b* = 0.05, *SE* = 0.03, 95% CI [−0.000, 0.109], *p* = 0.051).

Finally, and most importantly for our analysis, the indirect (and conditional) effects of loss of significance on intentions to use doping via the predominance obsessive passion and moral disengagement strategies were significant and positive at both levels of normative influence conditions. However, this indirect effect was greater in the high (1 *SD* above the mean) normative influence condition (Indirect effect = 0.02, BootSE = 0.002, 95% CI: [0.007, 0.027]) than in the low (1 *SD* below the mean) normative influence condition (Indirect effect = 0.005, BootSE = 0.002, 95% CI: [0.001, 0.010]). In summary, the proposed serial moderated mediation model was supported: Loss of personal significance made athletes more prone to doping intentions through a greater predominance of obsessive (vs. harmonious) passion towards their sporting activity, which, in turn, facilitated the adoption of moral disengagement strategies; interestingly, the latter effect was stronger when they perceived that significant others from their social network approved their choice.

## 4. Discussion

The understanding of why and how athletes would use doping substances merits immediate attention given the relative diffusion of this phenomenon [1,2,3]. Previous studies have suggested that the motives behind doping use can be diverse such as enhancing performance, appearance, social acceptance, and even achievement of fame and glory [19,20]. Moving from here, and basing our analysis on the conceptual framework proposed by Kruglanski and colleagues in explaining the motivational forces of radical and deviant behaviors [6,7,8], we propose that behind the intentions to use doping, intended as a deviant and counterfinal behavior [5,23,27], is the motivation to (re)gain personal significance (i.e., the need to matter, to count and be respected) prompted by experiences of significance loss [6]. We propose that for individuals seeking to restore personal significance, the doping possibility is facilitated by their enhanced obsession with their sportive activities, which, in turn, augments the facility to adopting moral disengagement strategies to find justifications for it, especially when they perceive that significant others of their social network (friends, family, acquaintances, etc.) would approve and encourage such behavior (i.e., when normative influence promote the choice to use PES).

Results of our survey conducted on non-professional athletes confirm our hypotheses. At the basis of our hypotheses is that the need for personal significance can motivate the choice of *counterfinal means* (i.e., doping) that is harmful but perceived as effective in achieving one’s goal (i.e., enhancing performance or appearance, and, thus, restoring significance).

As our data suggest, the stronger the lack of personal significance, the higher is the obsessive passion, and the use of moral disengagement mechanisms, that, in turn, are positively related to the intention to use doping.

These results are consistent with the previous ones framed in the Quest for Significance Theory [26], where the frustration of an existential need (i.e., significance) is related to a higher probability of using extreme means to restore it. In our study, the willingness to use doping was allowed by an obsessive involvement in one’s sport which, in turn, predisposes to the loosening of one’s moral handbrake, especially when significant others were perceived to approve and encourage doping usage. The positive relationships found between the predominance of obsessive (over harmonious) passion and moral disengagement and between the latter and doping intentions are all consistent with previous literature [40,49], as well as the moderating role of subjective norms [2]. Moreover, as far as we know, this is the first work that provides empirical evidence on extreme behaviors in sports within the theoretical framework of the Quest for Significance.

Nevertheless, we should acknowledge some limitations. Our sample was represented exclusively by university students (i.e., non-professional athletes), thus it would be useful to extend the analysis to professional athletes in competitive sports. We believe that our theory could actually also predict doping use in professional athletes searching for glory and fame through victory achievements. Future studies might profitably extend the present analysis to them. Additionally, since the instructions given to the participants included explicit references to sport and doping substances, their responses may have been biased by social desirability or other self-presentation motives. Therefore, the use of implicit measures, catching up more automatic, and thus, probably, more sincere responses, is also requested in future studies aimed at investigating this issue. This relates to another point linked with the fact that our study relied on self-report data; i.e., we are aware of the possible effect of the method variance on the significance of the results and their generalizability. We also need to be cautious about inferring causality to the relationships found due to the cross-sectional nature of the study. These limitations can be addressed by using experimental (e.g., experimentally manipulating the loss of personal significance) or longitudinal designs. Further research could thus implement longitudinal models to examine both whether the observed variables affect intentions over time and whether they affect people’s later use of doping substances.

Despite these limitations, we believe that the theoretical and applied value of this work lies in investigating the deep underlying motivations behind the intention to use doping substances.

## 5. Conclusions

The practical implications are related to the well-known health risks associated with PES [54] and its legal and career consequences—as the use of these substances in sport is considered unethical and risky for health and hence prohibited. Here we certainly do not exhaust the reasons why people might be willing to use doping substances; however, this study provided new insight into the motivational basis of their intentions. We specifically found that the mechanisms (i.e., predominance in the obsessive passion, moral disengagement) leading to these intentions follow a feeling of lack of personal significance. This is one of the factors that motivate various extreme and deviant behaviors up to self-sacrifice [22]. Consistently, helping people work on their personal significance by raising awareness of alternative ways to restore it could be key interventions to prevent or stem doping intentions and behaviors. As the need for mattering is a universal need, people who feel the lack can fill it in many positive and prosocial ways, and sport can generally be understood as one of them. When this turns into the choice of a counterfinal means, it can be harmful to the person [55]. These findings have important implications for the design of intervention studies aimed at preventing the use of doping substances by athletes and suggest that they could be helped to act on their personal significance, their sense of competition and success (preventing obsessive passion), and learn about the negative consequences of doping. This approach can also be used to develop targeted interventions (e.g., in the education field) aimed at raising awareness on the possible reasons for the use of drugs in sport and highlighting in advance, where possible, the availability of alternative means to satisfy salient basic needs. Communication to young people can also focus on fair play values (fair competition, sport without doping, integrity, etc.) as well as the importance of hard work and dedication, which could effectively contribute to that personal significance that people need. Consistent with this, the association between personal value (goal) and fair sporting practice (means) can be strengthened by focusing on the fact that sporting success through hard work is attributable to the person (i.e., more personal worth); instead, the association between personal value and doping can be weakened by focusing on the fact that sporting success with doping is attributable to external factors (i.e., substances) rather than to the person (i.e., less personal worth).

To conclude, we have shown that doping, generally seen as instrumental in achieving the goals of winning and/or appearance, is a more in-depth means of restoring or gaining personal significance. Future research could examine the possibility, as suggested in other contexts [50], that providing alternative means for restoration could reduce the attractiveness of more extreme means by diluting their instrumentality for goal achievement [56]. Given the breadth and resonance of the doping phenomenon, it is worth researching to explore the deeper grounds that motivate this behavior in order to attempt multiple ways to counter it.

## Figures and Tables

**Figure 1 ijerph-18-05411-f001:**
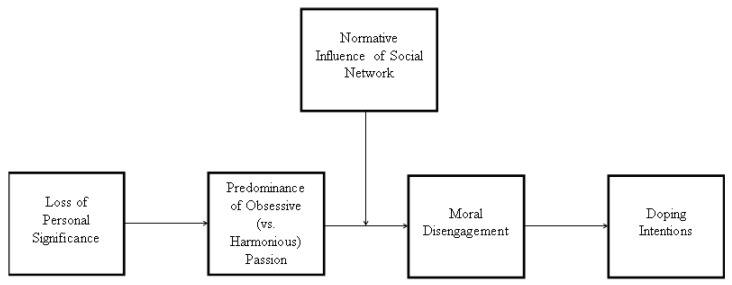
Graphic of the hypothesized serial moderated mediation model.

**Figure 2 ijerph-18-05411-f002:**
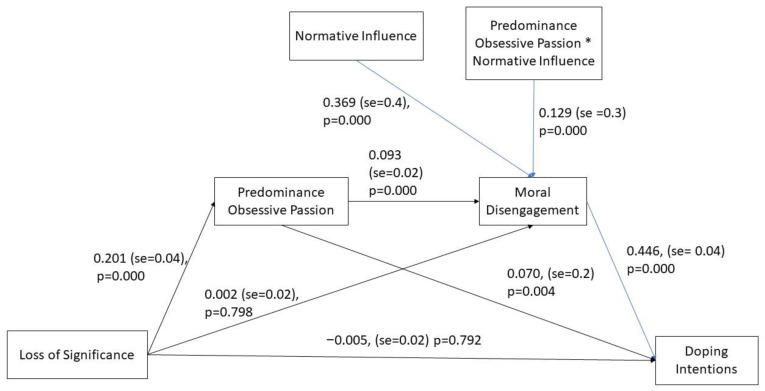
Results of the serial moderated mediation model.

**Table 1 ijerph-18-05411-t001:** Descriptive statistics and correlations among key variables of the model.

Variable	*M*	*SD*	1	2	3	4
1. Loss of Significance	2.88	1.39				
2. Predominance of Obsessive Passion	−1.44	1.27	0.219 **			
3. Moral Disengagement	1.58	0.73	−0.034	0.225 **		
4. Normative Influence	1.29	0.68	−0.030	0.122 *	0.408 **	
5. Intention to use doping	1.38	0.71	0.032	0.226 **	0.483 **	0.560 **

*Note*. *M* = Mean; *SD* = Standard Deviation; * *p* < 0.05; ** *p* < 0.01.

## Data Availability

The data presented in this study are available on request from the corresponding author.

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
