# Peer review of "The Motivational Underpinnings of Intentions to Use Doping in Sport: A Sample of Young Non-Professional Athletes"

_ijerph, 2021, doi:10.3390/ijerph18105411_

Round 1
Reviewer 1 Report
This topic is quite interesting and it is a well-done conducted research.
However, a major issue should be improve. It is not possible to interpret the results because the statistical analysis section is not clear for the reader.
How do the authors evaluated the moderated mediation model? It is nothing in the method section about these procedures.
Withouth a clear description it is not possible to understand the results section and also the discussion.
Please, put efforts to improve it.
Author Response
-REVIEWER 1:
This topic is quite interesting and it is a well-done conducted research.
- Thank you for your positive comment, it is very appreciated
However, a major issue should be improve. It is not possible to interpret the results because the statistical analysis section is not clear for the reader.
How do the authors evaluated the moderated mediation model? It is nothing in the method section about these procedures
Withouth a clear description it is not possible to understand the results section and also the discussion.
- Thank you for your comment, we added a specific data analysis section trying to describe the serial mediation model, however these models are quite complex, so we put some efforts in describing in a simpler way.
Please, put efforts to improve it.
Thank you we hope that this version has been improved enough
Reviewer 2 Report
A more standard abstract is needed. First, please add the research design. Also provide some details of the samples, settings, data collection tool, and results (statistical results and p values).
In the introduction, please specify the gap in the knowledge motivating you to conduct this study. Is there any previous study on this topic?
Regarding the sampling, please mention what a sampling and recruitment method (practically) has been used. How many were available and how many did not participate.
There is a need to a subheading for data analysis method and description of its details.
Please separate the ethics section and bring all its details under a separte subheading.
At the begining of the results, the description of the demographic data for the samples is needed.
Your description of the results is pure statistical. Please provide some description along with them in terms of your research variables and aim, that could suits your readers who may not have enough knowldge of statistics.
Please furnish your conclusion with practical considerations of your findings in terms of education, research, practice, and policy-making.
Author Response
A more standard abstract is needed. First, please add the research design. Also provide some details of the samples, settings, data collection tool, and results (statistical results and p values).
- Thank you, according to your comment we fully revised the abstract.
In the introduction, please specify the gap in the knowledge motivating you to conduct this study. Is there any previous study on this topic?
- Thank you for pointing this out, according to your comment we specified this aspect in the introduction.
Regarding the sampling, please mention what a sampling and recruitment method (practically) has been used. How many were available and how many did not participate.
- Thank you for the feedback, we added some specification about the recruitment process and the participation rate in the method section.
There is a need to a subheading for data analysis method and description of its details.
- Thank you for the comment, we added a specific data analysis section trying to describe the serial mediation model, however these models are quite complex, so we put some efforts in describing it in a simpler way.
Please separate the ethics section and bring all its details under a separte subheading.
- Thank you for the feedback, a separate section has been added to the method section.
At the begining of the results, the description of the demographic data for the samples is needed.
Thank you, according with your comment, we moved the description of demographic data from the method section to the results section.
Your description of the results is pure statistical. Please provide some description along with them in terms of your research variables and aim, that could suits your readers who may not have enough knowldge of statistics.
- Thank you for the feedback, we provided a more descriptive short summary in the results section.
Please furnish your conclusion with practical considerations of your findings in terms of education, research, practice, and policy-making.
- Thank you for the feedback, we extended practical implications according to your suggestion.
Round 2
Reviewer 1 Report
The authors have improved a lot the manuscript. All my questions were addressed by the authors. Congrats.
Author Response
Thank you very much for your comments. They were very helpful to us.
Reviewer 2 Report
Line 232, under material and methods, please add a subheading as 'Research Design' and recognise the study design under it.
Author Response
Thank you for your comment, under the material and method section we added a "Research design" subheading and we specified the design of the study.